# Seeing the Wind:
# Visual Wind Speed Prediction with a Coupled Convolutional and Recurrent Neural Network

**Jennifer L. Cardona**
Department of Mechanical Engineering
Stanford University
Stanford, CA 94305
jcard27@stanford.edu

**Michael F. Howland**
Department of Mechanical Engineering
Stanford University
Stanford, CA 94305
mhowland@stanford.edu

**John O. Dabiri**
Graduate Aerospace Laboratories (GALCIT)
and Mechanical Engineering
California Institute of Technology
Pasadena, CA 91125
jodabiri@caltech.edu

## Abstract

Wind energy resource quantification, air pollution monitoring, and weather forecasting all rely on rapid, accurate measurement of local wind conditions. Visual observations of the effects of wind—the swaying of trees and flapping of flags, for example—encode information regarding local wind conditions that can potentially be leveraged for visual anemometry that is inexpensive and ubiquitous. Here, we demonstrate a coupled convolutional neural network and recurrent neural network architecture that extracts the wind speed encoded in visually recorded flow-structure interactions of a flag and tree in naturally occurring wind. Predictions for wind speeds ranging from 0.75-11 m/s showed agreement with measurements from a cup anemometer on site, with a root-mean-squared error approaching the natural wind speed variability due to atmospheric turbulence. Generalizability of the network was demonstrated by successful prediction of wind speed based on recordings of other flags in the field and in a controlled wind tunnel test. Furthermore, physics-based scaling of the flapping dynamics accurately predicts the dependence of the network performance on the video frame rate and duration.

## 1 Introduction

The ability to accurately measure wind speeds is important across several applications including locating optimal sites for wind turbines, estimating pollution dispersion, and storm tracking. Currently, taking these measurements requires installing a physical instrument at the exact location of interest, which can be cost prohibitive and in some cases unfeasible. Knowledge of the wind resource in cities is of particular interest as urbanization draws a larger portion of the world's population to such areas, driving the need for more distributed energy generation closer to densely populated regions [20]. There is also burgeoning interest in the use of drones for delivery, which would greatly benefit from instantaneous knowledge of the local wind conditions to minimize energy consumption and ensure safety [34]. Here we demonstrate a technique that enables wind speed measurements to be made from a video of a flapping flag or swaying tree. This facilitates visual anemometry using pre-existing features in an environment, which would be non-intrusive and cost effective for wind mapping.

The flow-structure interaction between an object and the wind encodes information about the wind speed. Neural networks can potentially be used to decode this information. Here, we leverage machine learning to predict wind speeds based on these interactions. The general approach to the current problem uses a convolutional neural network (CNN) as a feature extractor on each frame in a sequence, followed by a recurrent neural network (RNN) taking in the features extracted from the time series of frames. The input to our algorithm is a two-second video clip (a sequence of 30 images), and the output is a wind speed prediction in meters per second (m/s).

This visual anemometry technique has the potential to significantly decrease the cost and time required for mapping wind resources. Installing an anemometer to monitor a single location typically costs $O(\$1,000)$, and even then only offers measurements at one location. While we have installed flags and trees at a field site to collect initial training and test data, the application of this method would occur using pre-existing structures in the environment of interest. Therefore, the only cost of this method is in the camera recording device (a standard camera phone provides sufficient resolution). Hence, the cost of this method is dramatically lower per measurement point, and the barrier posed by the time and labor required to install an anemometer at a location is removed.

## 2    Related Work

The innovation proposed in this study is to use videos observing flow-structure interactions to make wind speed predictions without using any classical meteorological measurements as inputs. This enables wind speed prediction in a much broader range of physical environments, especially those where meteorological sensors would be expensive or impractical to install. Classic methods of using visual cues to estimate wind speeds include the Beaufort scale, which provides a rough estimate based on human perception of the surrounding environment, and the Griggs-Putnam Index [36], which relies on the angle of plant growth to estimate annual average speeds. With recent advances in machine learning, the present work seeks to extend the capabilities of visual wind speed measurement to provide automated and quantitative real-time measurements.

Extracting physical quantities from videos has become increasingly prevalent. Several studies have used images or videos to estimate material properties of objects [37, 25], and specifically for cloth [6, 11, 39]. Video inputs have also been used to predict dynamics of objects [27], and physical properties of fluid flows [33, 30]. Estimating model parameters for physical simulations using similarity comparisons to video data has also shown promise in approximating static and dynamic properties of hanging cloth [5], the masses of colliding objects [38], and most recently, wind velocity and material properties given a flapping flag [29]. The success in this type of parameter estimation speaks to the potential for computer vision to be used in determining physical quantities.

There has long been interest in the use of neural networks in predicting future wind speeds based on historical measurements [26]. Much of the work in this area has been focused on wind forecasting using time series of measurements from existing instrumentation or weather forecast data as inputs [4, 23, 7, 13, 3, 9, 24]. Deep learning has recently been employed for meteorological precipitation nowcasting, where the authors used spatiotemporal radar echo data to make short-term rainfall intensity predictions using a convolutional LSTM [31].

Recent work has shown success in classifying actions and motion in video clips using deep networks. The present work aimed at wind speed regression from videos draws inspiration from previous studies aimed at classifying videos. In the deep learning era, a number of approaches to video classification have arisen using deep networks. Three notable strategies that have taken hold include 3D convolutional networks over short clips [2, 19, 22, 35], two-stream networks aiming to extract context and motion information from separate streams [32], and the combination of 2D convolutional networks with subsequent recurrent layers to treat multiple video frames as a time series [12, 41]. Carreia *et al.* performed a comparison of these different approaches to video classification [8]. This study will employ a strategy using a 2D CNN followed by long short-term memory (LSTM) layers. This approach leverages transfer learning on a CNN to extract features related to the instantaneous state of a flag, and a RNN to analyze the wind-induced flapping motion. Further discussion of this architecture choice can be found in Section 4.

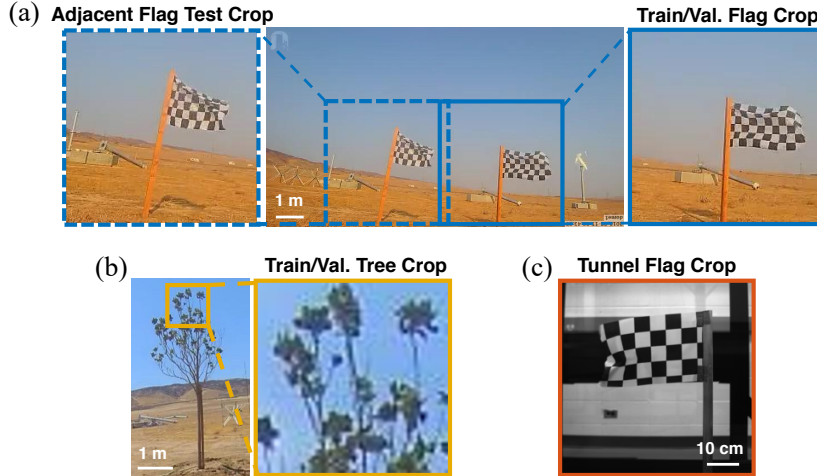

Figure 1: Examples of cropped video frame inputs for (a) the training/validation flag and adjacent test flag (b) the training/validation tree, and (c) the tunnel test flag.

## 3 Dataset

The main dataset used for training and validation consisted of videos taken at a field site in Lancaster, CA over the course of 20 days during August 2018. Only videos between the hours of 7:00 a.m. and 6:00 p.m. were used in order to ensure daylight conditions. Videos captured the motion of a standard checkerboard flag with an aspect ratio of 5:3 and size of 1.5 m × 0.9 m mounted at 3 m height (Figure 1a), as well as the canopy of a young southern magnolia tree (*Magnolia grandiflora*) of approximately 5 m height (Figure 1b). Videos were recorded at 15 frames per second. The videos were subsequently separated into two-second clips (30 sequential frames), which were formatted as RGB images cropped to $224 \times 224$ pixels. Thus, each sample used as a model input consists of a time series of images of total size $30 \times 224 \times 224 \times 3$. Ground truth 1-minute average wind speed labels were provided by an anemometer on site at 10 m height. The 1-minute averaging time was chosen instead of a shorter averaging time because of the highly turbulent and variable flow. The anemometer and flag are spatially separated, so the instantaneous measurements made by the anemometer do not correspond to exact instantaneous speeds experienced by the flag. Each image sequence was matched to a wind speed label using the timestamp of the first image in the series.

The measured 1-minute averaged wind speeds ranged from 0-15.5 m/s. The natural distribution of wind speeds was not uniform over the time-period of data collection, with many more samples in the middle speed ranges than at the tails. Since the desired output of the model was predictions over a broad range of wind speeds, a more uniform distribution was preferable. To achieve this, the ground truth wind speeds were binned in 0.25 m/s increments, and for each bin, excess samples were excluded to retain a more even distribution over the range of wind speeds. Clips were chosen at random for the training/validation split. The resulting training set contained $13,365$ clips ($10,490$ flag clips and $2,875$ tree clips). The validation sets for the flag and tree contained $4,091$ and $2,875$ clips respectively. Wind speed distributions for each dataset are provided in the Supplementary Materials document.

To asses the generalizability of the network, two test sets were collected containing videos of new flags in additional locations: a field test set (called the adjacent flag test set), and a wind tunnel test set (called the tunnel test set).

**Adjacent Flag Test Set**   The adjacent flag test set comprised clips of a flag identical to the one used for training/validation. The test flag was located 3 m from the training/validation flag (Figure 1c). Clips were taken at the same time as clips from the validation set to allow for direct comparison between validation results and test results on this new flag. Although the timestamps and wind speed labels are identical between the validation set and the adjacent flag test set, the precise wind

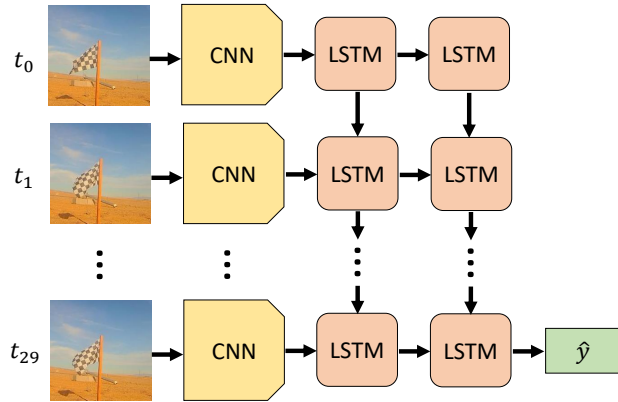

Figure 2: Schematic of the model architecture. The CNN is an pre-trained ResNet-18 architecture [14]. The LSTM is a many-to-one architecture with 2 layers each containing 1000 hidden units.

conditions and corresponding flag motion differ between the two, as turbulence is chaotic and variable in space.

**Tunnel Test Set**  The tunnel test set consists of clips taken of a smaller checkered flag mounted in a wind tunnel (Figure 1d). The tunnel flag had the same 5:3 aspect ratio as the other two flags discussed, but was 0.37 m $\times$ 0.22 m in size. The wind tunnel was run at three speeds: $4.46 \pm 0.45$ m/s , $5.64 \pm 0.45$ m/s, $6.58 \pm 0.45$ m/s. At each speed, $600$ two-second clips were recorded at 15 frames per second, yielding $1,800$ tunnel test samples. Although these videos were recorded on a monochrome camera, they were converted to 3-channel images by repeating the grayscale pixel values for each of the three channels for use in the model.

The final datasets used in this work are available at `https://purl.stanford.edu/ph326kh0190`.

## 4   Methods

### 4.1   Feature Extraction with ResNet-18

Before analyzing a video clip as a time series, each individual $224 \times 224 \times 3$ frame was fed through a 2D CNN to extract relevant features. The ResNet-18 architecture was chosen for the CNN because of its proven accuracy on previous tasks and relatively low computational cost [14]. Pre-trained weights for ResNet-18 were used in the current implementation to take advantage of transfer learning, available through the MathWorks Deep Learning Toolbox [18].

Since the purpose of the CNN here is feature extraction rather than image classification, the last two layers of the ResNet-18 (the fully connected layer and the softmax output layer) were removed so that the resulting output for each frame was a $7 \times 7 \times 512$ feature map. Since the activation function for the final layer was a rectified linear unit ($\text{ReLU}(x) = \max(0, x)$), many of the resulting features were zero. Therefore, to reduce memory constraints, the output features were fed through an additional maximum pooling layer with a filter size of 3 and a stride of 2. This acts to downsample the features and reduces the number of zeros in the dataset, reducing the feature map size to $3 \times 3 \times 512$ for each image, which was then flattened to $4,608 \times 1$. This resulted in a $4,608 \times 30$ output for each two-second (30 frame) clip to be used as in input for the recurrent network.

### 4.2   LSTM With and Without Mean Subtracted Inputs

A RNN was selected in order to learn temporal features of the videos. The flapping of flags is broadband, containing a wide range of spectral scales [1]. Typically, flags are located in the turbulent atmospheric boundary layer, where the length scales which govern the flow vary from the order of kilometers to the order of micrometers. As a result, the associated time scales will range from

Table 1: Final hyperparameter choices for LSTM networks.

| Hyperparameter | Chosen Value |
|---|---|
| # LSTM layers | 2 |
| # hidden units per LSTM layer | 1000 |
| learning rate | 0.01 |

milliseconds to minutes. Therefore, the architecture chosen for this application should adapt to the variable spectral composition of the flow field, which is captured by the motion of the flapping flag. The long short-term memory (LSTM) network was chosen for this application. It has been shown that the LSTM has the capability to learn interactions over a range of scales in a series [21], as well as advantages in training over longer time series [16], making it a suitable choice for this application.

A generic LSTM cell is computed with the input, forget, output, and gate gates:

$$
\begin{bmatrix} i \\ f \\ o \\ g \end{bmatrix} = \begin{bmatrix} \sigma \\ \sigma \\ \sigma \\ \tanh \end{bmatrix} W \begin{bmatrix} h_{t-1} \\ x_t \end{bmatrix},
\tag{1}
$$

and the cell and hidden states are computed as Equations 2 and 3 respectively [16]:

$$
c_t = f \odot c_{t-1} + i \odot g
\tag{2}
$$

$$
h_t = o \odot \tanh(c_t).
\tag{3}
$$

The weight matrix $W$ contains the learnable parameters. The sigmoid function, $\sigma$, is given by $\sigma(x) = e^x/(e^x + 1)$, and $\tanh(x) = (e^x - e^{-x})/(e^x + e^{-x})$. The LSTM is more easily trained on long sequences compared to standard RNNs because it is not susceptible to the problem of vanishing gradients, which arises due to successive multiplication by $W$ during backpropagation through a standard RNN [15]. In a LSTM network, the cell state allows for uninterrupted gradient flow between memory cells during backpropagation, as it requires only multiplication by $f$ rather than by $W$.

As discussed in Section 4.1, the inputs to the LSTM network are obtained from the features extracted from the pre-trained ResNet-18 network. Since wind conditions are influenced by the diurnal cycle and other weather conditions [10], it is plausible that a model could use features present in the video clips other than the motion of the objects (e.g. the position of the sun, presence of clouds). To study and avoid such artifacts from over-fitting to background conditions, two experiments were run using the same network architecture and hyperparameters, but trained using different inputs, referred to as LSTM-NM (short for no-mean) and LSTM-raw respectively.

**LSTM-NM**  In this experiment, the temporal mean of each feature over the 30-frame clip was subtracted from the inputs to avoid fitting to background features. These mean-subtracted feature maps served as inputs for the LSTM network.

**LSTM-raw**  Here, a second model was trained using the raw features extracted from the ResNet-18 model without mean subtraction. The main purpose of this experiment is to identify whether removing the temporal mean from a sequence is beneficial for model generalizability to new locations, and to confirm that it is in fact the object motion that is used for predictions.

The LSTM architecture used here is many-to-one, since we have a series of 30 images being fed into the LSTM network with only one regression prediction being made. A schematic of the overall architecture is shown in Figure 2. Hyperparameters were chosen based on values used for other spatiotemporal tasks with a similar model architecture in the literature [40, 41, 12]. The final size of the LSTM network was chosen to be 2 layers with 1000 hidden units per layer. Two smaller models were also considered (1 layer with 10 hidden units, and 1 layer with 100 hidden units), but these models suffered from high bias, and were under-fitting the training set. A summary of the chosen hyperparameters is shown in Table 1.

### 4.3 Implementation Details

This problem is framed as a regression, with a regression output layer that allows for the model to predict any wind speed as opposed to a specific class. The mean-squared error was used as the loss function, defined as:

$$L = \frac{\sum_i^N (y_i - \hat{y}_i)^2}{N},\tag{4}$$

where $N$ is the number of training examples, $y_i$ is the wind speed label, and $\hat{y}_i$ is the predicted wind speed label for the given training example. Mean-squared error was chosen over mean absolute error to more heavily penalize outliers, which are particularly undesirable in applications related to wind energy due to the cubic dependence of wind power on wind speed.

Stochastic gradient descent with momentum was used for optimization, with a typical momentum parameter of 0.9 [28]. The algorithm was implemented using the MATLAB Deep Learning Toolbox [17]. The LSTM network was trained for 20 epochs using minibatches of 256 samples. This amount of training was sufficient to over-fit to the training set within the limit of natural wind speed variability due to turbulence. Early stopping was employed for regularization. Computations were performed on a single CPU.

## 5 Results and Discussion

Table 2: Error metrics for validation and test cases. 'Overall RMSE' indicates the RMSE over all wind speeds, and 'Measurable Range RMSE' refers to the RMSE for wind speeds ranging from 0.75-11 m/s (described in Section 5.1.1).

| | LSTM-raw RSME (m/s) | | LSTM-NM RMSE (m/s) | |
|---|---|---|---|---|
| Dataset | Overall | Measurable Range | Overall | Measurable Range |
| Flag Validation Set | 1.37 | 1.27 | 1.42 | **1.37** |
| Tree Validation Set | 1.53 | 1.29 | 1.63 | **1.47** |
| Adjacent Flag Test Set | 2.77 | 3.10 | 3.02 | **1.85** |
| Tunnel Test Set | N/A | 9.61 | N/A | **1.82** |

### 5.1 LSTM-NM Validation Results

Figure 3a shows the mean wind speed predictions for 1 m/s bins plotted against the true 1-minute average wind speed labels for the validation set. The vertical error bars represent the standard deviation of predictions for each bin. The horizontal error bars show the range of wind speeds captured by each bin for the field datasets, and the accuracy of the anemometer measurements for the tunnel set. The overall root-mean-squared error (RMSE) for the flag and tree validation set were 1.42 m/s and 1.63 m/s respectively, which approach the natural wind variability due to atmospheric turbulence as will be discussed in Section 5.1.2. The mean prediction for each bin shows good agreement with the true labels, but the model tends to under-predict for high wind speeds ($\overline{U} > 11$ m/s), and over-predict for low wind speeds ($\overline{U} < 2$ m/s). Although some of error at low wind speeds might be due to a lack of training examples in that range (see the training distribution in the Supplementary Materials document), as discussed in detail in the next section, we found that the reduced accuracy at the lowest and highest wind speeds could be predicted based on knowledge of the physics of the flow-structure interaction, as well as the video sample duration and temporal resolution.

#### 5.1.1 Measurement Limitations at High and Low Wind Speeds

At the lowest and highest wind speeds tested, the increased prediction error can be explained by an inability of the current dataset to capture the relevant physics necessary to measure the wind speed.

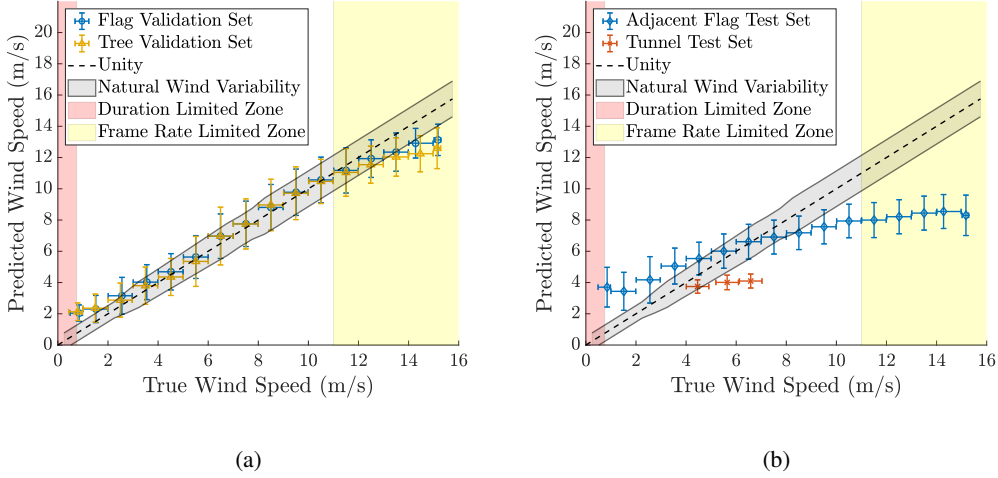

(a)                                                             (b)

Figure 3: Mean LSTM-NM model predictions as a function of the true wind speed label for (a) the validation set and (b) the test sets. A perfect model would carry a one-to-one ratio, indicated by the 'Unity' line overlaid on the plot (dashed black line). Vertical error bars indicate one standard deviation. Horizontal error bars indicate the range of wind speeds represented by a mark based on the binning for the field datasets, and the measurement uncertainty from the anemometer used in the the tunnel test set. The wind speeds outside of the measurable range due to clip duration and frame rate are shown by the shaded red and yellow shaded regions respectively.

Here we will look more specifically at the flapping flag in the field for illustration. The pertinent physics are captured by a frequency scale of order $f$ where,

$$f = U/L \tag{5}$$

is the frequency corresponding to a fluid element passing by the flag, $L$ is the length of the flag, and $U$ is the wind speed.

At high wind speeds the measurement capabilities are limited by the sampling rate, $f_s$. The Nyquist frequency, defined as $f_{Nyquist} = 0.5 f_s$, is the highest frequency that a signal can have and still be observed without the effects of aliasing. Using the Nyquist frequency as an upper bound for the characteristic frequency, the corresponding critical velocity, $U_{c,high}$ can be calculated as:

$$U_{c,high} = L f_{Nyquist} \tag{6}$$

In this case, $f_s = 15$ Hz, given by the frame rate, yields a Nyquist frequency of $f_{Nyquist} = 7.5$ Hz. The length of the flag is fixed at $L = 1.5$ m. Applying these values to Equation 6 gives $U_{c,high} = 11$ m/s. For wind speeds exceeding this value, the characteristic frequency would not be measurable without aliasing. This appears to manifest as an under-prediction at high wind speed values in Figure 3a.

At low wind speeds, the duration of clips, $T$, is the limiting factor. The lowest frequency that can be fully observed is $f = 1/T$. The critical velocity is then given by:

$$U_{c,low} = L/T \tag{7}$$

Given a clip length of 2 seconds, $U_{c,low} = 0.75$ m/s. Wind speeds lower than that would have fundamental frequencies that are too low to fully observe. Because of these known limitations for model performance at speeds under 0.75 m/s and above 11 m/s, the RMSE for range 0.75-11 m/s (hereafter referred to as the measurable wind speed range) has been reported in addition to the overall RMSE. For the flag validation set, the RMSE within this range was 1.37 m/s (results summarized

in Table 2). The red and yellow shaded regions in Figure 3 indicate wind speeds outside of this measurable range due to duration and sampling rate respectively.

### 5.1.2 Comparison to Turbulent Fluctuations

In evaluating model performance, it is important to consider the natural variation in the wind speed due to turbulence. Because of this variation, it is expected that the RMSE for the model predictions is at least as large as the standard deviation of turbulence fluctuations (denoted $\sigma_u$) calculated over the 1-minute averaging time. The fluctuating velocity, $u'$, and $\sigma_u$ are given in Equations 8 and 9 respectively:

$$u' = u(t) - \overline{U} \tag{8}$$

$$\sigma_u = \sqrt{\overline{u'^2}} \tag{9}$$

where $u(t)$ is the instantaneous velocity, and $\overline{U}$ is the time-averaged velocity. To calculate $\sigma_u$ for our field site, 1-minute average wind speed measurements were used for the mean velocity, $\overline{U}$, and 2-second averages were used to represent the instantaneous velocity, $u(t)$. The 2-second averaging time for the instantaneous measurements was chosen in order to match the duration of the video clips used as model inputs. Thus, each prediction from the network can be seen as a comparable instantaneous measurement, and in the ideal case, the standard deviation of the predicted values should match the standard deviation of the instantaneous anemometer measurements at each wind speed. To determine $\sigma_u$ over the range of wind speeds, measurements of $\overline{U}$ were binned in 0.5 m/s increments, and the mean natural variability due to turbulence ($\sigma_u$) was calculated for each bin, represented by the gray band in Figure 3a.

This analysis allows for two comparisons of predictions to anemometer data. The first comparison is to 1-minute average wind speeds that represent $\overline{U}$ at the site at a given time, which is shown by the markers in Figure 3a compared to the dashed black line representing unity, which show good agreement. The second comparison is between the standard deviation of predictions to the $\sigma_u$. The size of error bars representing the standard deviation in predictions is approaching the size of the gray band shown for $\sigma_u$. This result indicates that the model performance approaches the best possible accuracy given natural wind variability.

### 5.2 LSTM-NM Test Results and Model Generalizability

Model predictions for both the adjacent flag test set and the tunnel test set are plotted against true labels in Figure 3b. As discussed in Section 3, the adjacent flag test set serves as a direct comparison to the validation set. Although the model still captures the increasing trend, the over-predictions at high wind speeds and under-predictions at low wind speeds are more pronounced than they were for the validation set, visible in the flatter shape of the curve shown in Figure 3b.

The tunnel test set results are shown by the orange marks in Figure 3b. Similarly to the adjacent flag test set, the predictions for the tunnel test set capture the correct qualitative increasing trend, although the highest wind speed case appears to be under-predicted.

There are plausible explanations for why the test set predictions lie in a narrower range than the validation set predictions. For the tunnel test set, the flag length is shorter (0.37 m), which means the model may be limited by physics at even lower speeds (Section 5.1.1). The narrower range of predictions for the corrected adjacent flag test set suggests that the model may be partially over-fit to the specific flag and tree it has been trained on (i.e. relying partly on specific features of those objects). The effect of over-fitting may become less significant if the training set were expanded to include a more diverse set of flags and trees.

Test set predictions are still monotonically increasing with increasing true wind speed, suggesting that model capabilities are only partially limited by factors such as the frame rate or over-fitting. For both test sets, the RMSE in the measurable wind speed range was close to the validation set (Table 2). This indicates that the current model has potential to make predictions for flags other than the one it has been trained on, and flags that exist in new locations. This suggests the possibility for

generalizability of this type of model in new settings, and its potential for broader use in mapping wind speeds.

### 5.3 LSTM-raw Results and Effect of Mean Subtraction

As discussed in Section 4.2, the LSTM-raw experiment used the same model architecture and hyperparameters as the LSTM-NM experiment, but used raw inputs rather than inputs with temporal mean-subtracted inputs. The LSTM-raw model performed very similarly to the LSTM-NM model on the validation sets (Table 2). However, it performed notably worse on both test sets. For the adjacent flag test set, the LSTM-raw model gave a RMSE of 3.10 m/s in the measureable range compared to 1.85 m/s for the LSTM-NM model. For the tunnel test set, the LSTM-raw model gave a RMSE of 9.61 m/s, compared to 1.82 m/s for LSTM-NM. The decrease in performance on the test sets for the LSTM-raw model indicates that without a temporal mean subtraction, this model was unable to make accurate predictions for a flag in a new location with a different background.

## 6 Conclusion and Future Work

Here, a coupled CNN and RNN using the ResNet-18 and LSTM architectures was trained to successfully predict wind speeds within a range from 0.75-11 m/s using videos of flapping flags and swaying trees, with prediction errors approaching the minimum expected error due to turbulence fluctuations on the validation set. Model performance on test sets consisting of new flags in additional locations suggests that such a model may generalize, and could therefore prove useful in measuring wind speeds in new environments. Using this data-driven approach to visual anemometry could offer significant benefits in applications such as mapping complex wind fields in urban environments, as it could cut down on the time and cost required to measure wind speeds at several locations, which is currently done by installing an instrument at each point of interest.

Although this study focused specifically on video clips of checkered flags and a magnolia tree, this approach to wind speed measurement can potentially generalize to other types of objects, including other types of flags and natural vegetation that interact with the surrounding wind. In addition to training on a broader dataset including a variety of objects, confidence in the potential for model generalization can be further improved given a deeper understanding of which relevant physics the model is using for prediction. Here we showed how the measurement capabilities of a model were limited by the fundamental frequency of the flag. Future work will focus on understanding which physics of the fluid structure interactions are extracted by the model and are necessary for accurate predictions.

## 7 Acknowledgements

The authors acknowledge Kelyn Wood, who assisted in setup for the wind tunnel test set. J.L.C. is funded through the Brit and Alex d'Arbeloff Stanford Graduate Fellowship, and M.F.H. is funded through a National Science Foundation Graduate Research Fellowship under Grant DGE-1656518 and a Stanford Graduate Fellowship.

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
