[Reviews · NeurIPS 2019]

Reviewer 1



[Updated after the author feedback] =Summary= The goal of the paper is to devise a mechanism to predict wind speed based on the 2-second video snippets on a checkered flag. A resnet-18 is used as a feature extractor for each frame. The timeseries of features is then fed to an LSTM to predict the wind speed. A specialized dataset is also collected for the aforementioned task. The paper attempts an interesting problem. The idea of using visual signals to predict wind speed is indeed quite clever. However, in the current state, the paper needs more work before it is ready for publication. On the plus side, the collected dataset is quite useful for the boarder ML community. On the negative side, the paper neither introduces a theoretical or a methodological novelty, nor does it show striking empirical evidence. The empirical results are somewhat underwhelming and certain choices and intuitions are not explained very well. Overall, this is a promising research direction, and the authors are encouraged to pursue it by refining the writing and expanding the experimental setup. Please see the suggestions for improvements below. =Originality= Medium. The paper applies well-known models to a specific application. The gathered dataset however can prove to be quite useful. =Quality= Low. The model choices could be made in a more careful manner. For example, it is not clear why only recurrent networks would need to be used here. Please see detailed comments below. =Clarity= Medium. The content is generally accessible. =Significance= Low-Medium. The data collection and curation procedure used in the paper is quite thorough and could serve as a nice guideline for ML practitioners. However, on the negative side, the paper is very application specific and the results are somewhat underwhelming. Given the limited scope of the empirical validation, the paper is unlikely to have significant impact on the broader ML community. = Specific comments and suggestions for improvement = - The rebuttal was helpful in clarifying the operational mode of the proposed method, that is, once the method training is done, one can use the pre-existing structures e.g., trees and flags (as opposed to planting custom flags at each point of interest) to measure wind speed. This is a pretty clever idea and this reviewer highly recommends explicitly mentioning it in the intro. - In relation to the previous point, the performance on test sets (tunnel test set and adjacent flag test set) is a bit underwhelming. Looks like the method is overfitting to the main flag. Perhaps, it would help to have a separate validation flag and fine tune the model hyperparameters (architecture, early stopping) based on performance on this separate validation flag? Most importantly, it would be great to extend the experimental setup to other (non-checkered) flags/trees to empirically show that the idea has the promise to generalize. In summary, the paper can greatly benefit from a more comprehensive empirical section. - The intro mentions that the paper leverages both "physics and machine learning to predict wind speeds". However, the method is completely data-driven (CNN + LSTM). - It is not clear why the LSTM needs to be used in this setting. It is true that LSTMs are the first model that one thinks of when modeling timeseries data, however, for data with fixed length and relatively small (30 timesteps) trajectories, a CNN might work equally well. Additionally, it seems that given the nature of the problem (motion of an object over time), it seems that the self-attention mechanisms might be well suited for this task (https://arxiv.org/pdf/1711.07971.pdf). Regarding the answer in rebuttal to this point, this reviewer is not convinced that once can use the LSTM trained on 15 fps to make predictions on 30 fps. - It section 3, it seems like bins containing disproportionately large data are down-sampled. Rather than discarding data, why not upsample the sparse bins? - While this reviewer is not an expert on wind speed prediction, it sounds like taking 1-minute speed averages might be too noisy, that is, wind speed might change a lot during this time. It might be possible to check this variance from the data. The question that remains after the rebuttal is that how precisely was the number of 1-minute chosen? How would the takeaways change if one considered 30-second averages? - How were the training and validation sets separated? If one considers a 1-minute interval, then there are thirty consecutive 2-second intervals in this time period. However, all of these intervals might potentially have the same wind speed value (since it is a one minute average). Giving this overlap, would it make sense to split the training and validation data such that it does not overlap in these 1-minute intervals? - It would be great to mention if / how the analysis of 5.1.1 gets affected by the fact that the ground truth wind speeds considered in the paper are 1-minute averages.

Reviewer 2



This paper uses a coupled CNN and RNN architecture to predict wind speeds from 2-second clips of flags (specifically, observing their flow-structure interactions). The authors construct a dataset of checkered flags in three environments: (1) at a field site in Lancaster, CA, (2) at the same field site and with the same type of flag, but 3m away from the original flag, (3) in a wind tunnel, with a smaller-sized flag, and with monochrome camera videos. They then employ their model on this dataset, using a CNN to do feature extraction and then using an RNN to predict the wind speed associated with a 2 second (30 frame) flag video. The authors employ mean subtraction on their dataset to remove background features so that their model will better generalize to different conditions. They also contextualize the performance of their model at different wind speed ranges via frequency analysis (of the Nyquist frequency and clip duration-limited frequency) and turbulence analysis (to understand error bars). Originality: To the best of my knowledge, this is the first work that uses ML for wind speed prediction from flags. The closest related work (which the authors cite) is in extracting wind measurements from instrumentation (their instrument of choice here is nontraditional -- a flag) and in video prediction (which has not before been applied to this domain). From an applications perspective, this is therefore certainly a novel contribution (and one that would be useful to practitioners). The model evaluation portion also demonstrates originality, as model predictions are contextualized using frequency and turbulence equations. Quality: The work seems very well-executed, with assumptions and limitations both explained and contexualized in the domain. The model seems to predict wind speeds well on the main and adjacent flag test sets (i.e. the ones at the field site) for a variety of wind speeds, and does okay on the wind tunnel test set (which is not surprising given the different camera type and different flag size). I wish there had been more analysis of why the authors feel predictions for the tunnel test flag set were so uniform. I also have questions about the error bar calculations (see next section of the review). Overall, however, I believe this is a solid, well-executed contribution with good results, and that it poses a lot of challenges for future work to build on. Clarity: The submission is very clear and easy to follow. The description of the related work, dataset, method, model evaluation are all very clear and written in an engaging manner. Given the dataset, I could likely reproduce the authors' approach. Significance: I believe this work is significant from multiple perspectives. First, it provides a novel method for predicting wind speeds using commonly-available "instrumentation." Second, it provides a great case study of domain-specific considerations that are necessary for applying ML in the field -- for instance, previous ML algorithms have overfit to background features, and the authors explicitly account for this by collecting and using a diverse dataset. Third, it provides a dataset that the community can use for further exploration of this problem (though this dataset can and should be expanded upon with a more diverse set of flags and conditions in future work). EDITS AFTER DISCUSSION & AUTHOR FEEDBACK: * Initially, my biggest concern about this paper was that the dataset was not diverse (and I believe this concern was shared by the other reviewers). That said, the author response indicates that the authors have collected additional data (tree canopy movement), and that their model performs well on the tree canopy validation set when trained on tree canopy data. This additional data increases my impression of the value of the dataset presented here, and I do believe the dataset is one of the biggest contributions of this paper. * I am now less convinced that the model generalizes due to the flatness of the test set results (though it's worth noting that most NeurIPS papers evaluate on the equivalent of the validation set -- on which the authors' models perform well -- as opposed to the test set). (I also laud the authors for their honesty in providing corrected results.) Additionally, while I am glad to see that the model performs well on the tree canopy validation set when trained on tree canopies, to understand generalization, it would possibly be more important to see whether e.g. a model trained on the checkered flags generalizes to other flags/the tree, or a model trained on this tree generalizes to other trees. Given my increased impression of the dataset and my slightly decreased impression of generalization, my score remains the same.

Reviewer 3



UPDATED REVIEW: I appreciate the extra experiments and clear explanation of them, and am happy to raise my score. I would have liked to see some discussion of the cost of flags vs. anemometers, but maybe this is in the "specific comments" that the authors say will be incorporated. ================== Review summary: The paper is well-written, the dataset and the experiments done are well explained and careful. I like this paper and definitely want to encourage this line of work, but I am on the fence about whether there is sufficient experimentation here to merit publishing at this stage. Originality: The idea of estimating wind speed from imagery (video) is novel to my knowledge, and in my opinion is the main contribution of the paper. There is nothing novel about the model (CNN feature extractor with an RNN on top), but the authors don't claim there is, and based on the results it seems to perform well. Quality: The paper is well-written, and the dataset and experiments seem to be well described and of good quality. In order to be a really high quality paper though, I would want it to convince me that this idea has practical merit - i.e. at a minimum I would want to see experiments with one other video, e.g. a different type of flag, or a moving tree. Clarity: I found everything very clear; one of the strongest aspects of the paper. Significance: Difficult to judge without more convincing experiments, but it is at least an interesting idea and the dataset is a solid contribution; overall I would say "medium". - what about using auditory information and/or natural language reports of wind speed along /instead of the video? I would think this would be even cheaper than video, and making use of multimodal information would fit with the motivation to use 'existing' data in practical situations (e.g. drone delivery). This is just an idea, not a suggested improvement.

[Author Response · NeurIPS 2019]



**Figure 1:** a) Tree used for expanded dataset (sample crop region shown in white box); b) Example of tree canopy model input; c) Validation set results (horizontal error bars represent range of wind speeds within each bin); d) Test set results

**To all reviewers:** The primary question raised in the reviews was the extent to which the results would generalize beyond the checkered flags in the initial data set. To directly address this concern, we have retrained the model using videos of both checkered flags and a tree that we planted at the field site (Fig. 1a, 1b). The model performs well on the validation sets of the flag and the tree (Fig. 1c), demonstrating the potentially broad application of the presented method. We also corrected an error in the original adjacent flag test set, which results in a flatter profile, especially in the high wind speed range, where we expect the model to be Nyquist-limited by the frame rate (Fig. 1d). This correction does not change the conclusions of the paper.

**Reviewer 1:** While we do not claim a novel ML architecture, we believe that our work is appropriate for the Applications subject area at NeurIPS because we make a novel application of known techniques to an important problem. Installing an anemometer to monitor a single location costs 2000-3000 USD, and even then only offers measurements at one location. This application of ML potentially enables spatially resolved measurements within a full 2-D scene, as opposed to existing single-point anemometer measurements. To clarify, while we have installed flags and trees at a field site to collect initial training and test data, the application of this method would occur using pre-existing structures in the environment of interest, such as flags and trees. Therefore, the only cost of this method is in the recording device. A standard camera phone provides sufficient resolution, hence the cost of this method is dramatically lower per measurement point. We have also collected an unprecedented dataset, comprising over 50 hours of raw video data of flapping flags and trees in the field, which will be available for others to use. Although other model types could also be well suited for this task, we chose a CNN+LSTM model to leverage transfer learning and parameter sharing to minimize training time, especially because of the demanding data collection and pre-processing required for this work. The LSTM model also allows for different clip lengths to be used in future iterations, which may be important if different frame rates or durations are necessary to capture physics (Sec. 5.1.1). 1-minute averages were used for labels because of the highly turbulent and variable flow. The anemometer and flag are spatially separated, so the instantaneous measurements made by the anemometer do not correspond to exact instantaneous speeds experienced by the flag.

**Response to reviewer 2:** Vertical error bars represent one standard deviation of the mean prediction in each wind speed bin. This is used to estimate the model capabilities over the range of wind speeds. Horizontal error bars were added to show ranges of true speeds within each bin. For the turbulence fluctuation band, 2-second averages were chosen to match the clip length. Although the anemometer is situated higher than the flag, turbulence intensity typically decreases with height in the atmospheric boundary layer, so this band represents a conservative estimate for the fluctuations at the flag height. There are plausible explanations for why the test set predictions lie in a narrower range than the validation set predictions. For the tunnel test set, the flag length is shorter (0.37 m), which means the model may be limited by physics at even lower speeds (Section 5.1.1). In all sets, we still see monotonically increasing predictions in the frame rate limited zone, suggesting that frame rate only partially limits model capabilities. Predictions for the corrected adjacent flag test set are also flatter, which suggests that the model may be partially over-fit to the specific flag and tree it has been trained on (i.e. relying partly on specific features of those objects). The effect of overfitting will become less significant as additional field data is collected for the flags and tree training sets.

**Response to reviewer 3:** In response to the reviewer's comments, we have significantly expanded our training and validation sets to include tree data, which is qualitatively different from the original flag videos. Results suggest that the presented technique can indeed be extended to other objects given the additional data. Although wind speed labels are 1-minute averages, the problem is framed as a regression, with a regression output layer that allows for the model to predict any wind speed as opposed to a specific class. We quantify predictions using RMSE for given wind speed ranges in Table 2, and the error bars on the markers indicate bin-specific performance estimates. If accepted, the paper will be revised to include a more extensive literature review on wind speed estimation, and the reviewer's specific comments will be incorporated.

[Meta-Review · NeurIPS 2019]

The paper shows that accurate wind speed measurements in real time can be done using a suitable deep net based on visual observations such as flapping of flags or swaying of trees. The deep net considered is a coupled CNN and RNN. The results illustrate the approach to be accurate and discussions are provided for the challenges in the high and the low wind speeds, respectively called the frame rate limited zone and the duration limited zone. The reviewers agreed that the paper presents an interesting dataset and proposes a creative approach using existing machine learning models. The reviewers felt that due to the novelty of the application domain, novel machine learning approaches are not a requirement. However, opinion was divided on the overall merit of the work. Some reviewers felt that the paper makes progress on a problem which can have impact in real world whereas others felt that the paper does not solve a real world problem with a novel approach and has not illustrated the method to work well on test sets and in related settings. The work was extensively discussed after the author response but the difference in opinion could not be resolved. Adding a clear motivation for real world applications and discussions of the state-of-the-art will strengthen the paper.